# Tandem Suzuki Polymerization/Heck Cyclization Reaction to Form Ladder-Type 9,9′-Bifluorenylidene-Based Conjugated Polymer

**DOI:** 10.3390/polym15163360

**Published:** 2023-08-10

**Authors:** Xiaoyan Zhu, Feng Liu, Xinwu Ba, Yonggang Wu

**Affiliations:** 1College of Chemistry and Materials Science, Hebei University, Baoding 071002, China; zhuxiaoyan8910@126.com (X.Z.); baxw@hbu.edu.cn (X.B.); 2College of Basic Medicine, Hebei University, Baoding 071002, China

**Keywords:** 9,9′-bifluorenylidene, ladder type conjugated polymer, tandem reaction, Suzuki polymerization, Heck cyclization

## Abstract

The synthesis of ladder-type 9,9′-bifluorenylidene-based conjugated polymer is reported. Unlike the typical synthetic strategy, the new designed ladder-type conjugated polymer is achieved via tandem Suzuki polymerization/Heck cyclization reaction in one-pot. In the preparation process, Suzuki polymerization reaction occurred first and then the intramolecular Heck cyclization followed smoothly under the same catalyst Pd(PPh_3_)_4_. The model reaction proved that the introduction of iodine (I) for this tandem reaction can effectively control the sequential bond-forming process and inhibit the additional competitive side reactions. Thus, small-molecule model compounds could be obtained in high yields. The successes of the synthesized small molecule and polymer compounds indicate that the Pd-catalyzed tandem reaction may be an effective strategy for improving extended π-conjugated materials.

## 1. Introduction

Ladder-type conjugated polymers (LCPs) with an uninterrupted sequence of fused rings offer significant advantages over the traditional conjugated polymers [1,2,3,4,5]. The rigid structures of LCPs are beneficial for diminishing the torsional defects and improving intra-chain charge migration and extended exciton diffusion [6,7,8,9]. Thus, LCPs are endowed with extraordinary photoelectric properties and have become promising for a wide variety of applications, such as organic light-emitting diodes (OLEDs), organic solar cells (OSCs) and organic thin film transistors (OTFTs). Despite those great advances, the harsher polymerization conditions of LCPs have limited the extending application of LCPs and significant effort has been devoted to explore the polymerization conditions of LCPs [2,10,11,12,13].

Typically, LCPs have been synthesized via a multistep ladderization reaction, a first polymerization reaction and then a post-polymerization cyclization reaction [14,15,16,17]. This ladderization strategy enjoys a wider scope of reactions and monomers, but they may increase the byproducts or wastes, reaction time, and energy. The other strategy is a single-step ladderization reaction [18,19]. Although, the single-step process usually achieves well-defined backbones with minimum levels of structural defects, the polymeric monomers limits the application of this strategy.

A tandem reaction, which refers to combining multiple reactions in one-pot, can be regarded as an integration of the above two synthetic strategies [20,21]. The tandem reactions strategy may be one of multiple desirable processes for the polymerization of LCPs. Our group have made some attempts. We once successfully constructed ladder-type C-N-linked conjugated polymers via tandem reaction; specifically, we used the Pd-catalyzed Suzuki coupling/Schiff’s base reaction [22]. The model compounds are achieved in high yields. LCPs display desirable solubility and thermal stability. In addition, the Pd-catalyzed Suzuki coupling/Schiff’s base reaction, Pd-catalyzed Suzuki coupling/Heck cyclization reaction was also used as a tandem reaction to prepare new compounds with high stereoselectivity and yield [23,24,25].

9,9′-bifluorenylidene (99′BF), as one of the fulvalene derivatives, is a versatile scaffold with twelve different sites for functionalization by substitution. Though 99′BF-based compounds have been used as optoelectronic materials, such as an electron acceptor in bulky heterojunction solar cells, hole-transporting materials for perovskite solar cell and so on [26,27,28,29,30], the synthesis of 99’BF-based compounds requires extensive research and improvement due to its inherently crowded structure, which often results in low yield [31,32,33]. Meanwhile, the ladder-type 9,9′-bifluorenylidene-based conjugated polymers have not been reported. We have recently developed the synthetic methodology and successfully prepared ladder-type conjugated fulvalene oligomers via Pd-catalyzed Suzuki coupling/Heck cyclization reaction in high yields [34]. Drawing inspiration from this synthetic approach, we envisioned our blueprint for constructing the ladder-type 9,9′-bifluorenylidene-based conjugated polymer via tandem Suzuki polymerization/Heck cyclization reaction.

## 2. Synthesis Procedures

Synthesis of BFY2. A mixture of M2 (81 mg, 0.17 mmol), M5 (40 mg, 0.083 mmol) and NaHCO_3_ (0.30 g) was dissolved into THF (20 mL) and H_2_O (4 mL) under argon protection. After Pd(PPh_3_)_4_ (10 mg, 0.0087 mmol) was added into the mixture, the reaction mixture was stirred at reflux for 2 d. After cooling down, the reaction mixture was extracted with dichloromethane. The organic phase was dried over anhydrous Na_2_SO_4_ and filtered. After removing the solvent from filtrate, the crude product was purified by column chromatography on silica gel with petroleum ether/dichloromethane (10:1) to obtain the pure compound BFY2 as a purple solid (60 mg, yield 90%).

A mixture of M4 (80 mg, 0.089 mmol), M6 (40 mg, 0.199 mmol) and NaHCO_3_ (0.30 g) was dissolved into THF (20 mL) and H_2_O (4 mL) under argon protection. After Pd(PPh_3_)_4_ (10 mg, 0.0087 mmol) was added into the mixture, the reaction mixture was stirred at reflux for 2 d. After cooling down, the reaction mixture was extracted with dichloromethane. The organic phase was dried over anhydrous Na_2_SO_4_ and filtered. After removing the solvent from filtrate, the crude product was purified by column chromatography on silica gel with petroleum ether/dichloromethane (10:1) to obtain the pure compound BFY2 as a purple solid (66 mg, 93%): ^1^H NMR (CDCl_3_, 400 MHz): δ (ppm) 8.73 (s, 2H), 8.61 (s, 2H), 8.47 (s, 2H), 8.41 (d, J = 8.0 Hz, 2H), 7.63–7.56 (m, 6H), 7.42–7.35 (m, 4H), 7.28 (m, 2H), 7.18 (t, J = 7.6 Hz, 2H), 1.36 (s, 18H), 1.33 (s, 18H); ^13^C NMR (400 MHz, CDCl_3_): δ (ppm): 149.8, 149.7, 142.3, 141.3, 140.3, 140.0, 139.3, 139.2, 139.1, 138.5, 138.4, 129.1, 126.5, 126.5, 124.6, 124.5, 119.6, 119.4, 119.3, 118.2, 35.1, 35.1, 31.6, 31.4. MS (ESI) m/z: [M + H]^+^. calcd for C_62_H_59_ 803.4617; found 803.339.

Synthesis of Polymer LCPs1. A mixture of M3 (0.10 g, 0.12 mmol), M5 (54 mg, 0.11 mmol) and NaHCO_3_ (0.30 g) was dissolved into THF (20 mL) and H_2_O (4 mL) under argon protection. After Pd(PPh_3_)_4_ (8 mg, 0.0069 mmol) was added into the mixture, the reaction mixture was stirred at reflux for 3 d. M1 (15 mg, 0.034 mmol) and Pd(PPh_3_)_4_ (4 mg, 0.0035 mmol) were added into the reaction mixture under argon protection. The mixture was stirred at reflux for 2 d. After cooling down, the mixture was extracted with dichloromethane. The organic phase was dried over anhydrous Na_2_SO_4_ and filtered. After removing the solvent from filtrate, the crude polymer was precipitated from dichloromethane solution in methanol to give polymer LCPs1 as a deep red solid (61 mg, yield 76%): ^1^H NMR (CDCl_3_, 400 MHz): δ (ppm) 8.80–8.17 (broad, 3H), 7.84–7.30 (broad, 10H), 7.26–7.00 (broad, 3H), 1.56–0.85 (broad, 36H); ^13^C NMR (100 MHz, CDCl_3_): δ (ppm): 150.2, 149.8, 139.3, 139.0, 138.6, 137.1, 136.3, 135.5, 126.6, 126.4, 123.1, 122.7, 121.5, 119.4, 119.2, 117.5, 35.1, 34.9, 31.7, 31.5, 22.7, 14.2. Anal. calcd for C_56_H_52_: C 92.77, H 7.23; found: C 84.19, H 6.03, Br 3.26.

Synthesis of Polymer LCPs2. A mixture of M4 (0.15 g, 0.17 mmol), M5 (97 mg, 0.20 mmol) and NaHCO_3_ (0.30 g) was dissolved into THF (20 mL) and H_2_O (4 mL) under argon protection. After Pd(PPh_3_)_4_ (10 mg, 0.0087 mmol) was added into the mixture, the reaction mixture was stirred at reflux for 3 d. After cooling down, M1 (20 mg, 0.045 mmol) and Pd(PPh_3_)_4_ (4.0 mg, 0.0035 mmol) were added into the reaction mixture under argon protection. The mixture was stirred at reflux for 2 d. After cooling down, the reaction mixture was extracted with dichloromethane. The organic phase was dried over anhydrous Na_2_SO_4_ and filtered. After removing the solvent from filtrate, the crude polymer was precipitated from dichloromethane solution in methanol to give polymer LCPs2 as a black solid (0.11 g, yield 93%): ^1^H NMR (CDCl_3_, 400 MHz): δ (ppm) 8.69–8.43 (broad, 5H), 7.70–7.29 (broad, 10H), 7.20–7.18 (broad, 1H), 1.43–0.85 (broad, 36H). Anal. calcd for C_56_H_52_: C 92.77, H 7.23; found: C 85.36, H 6.46, Br 0.97.

## 3. Results

As shown in Figure 1, the model compound BFY2 was obtained via Pd(PPh_3_)_4_-catalyzed coupling reaction with M1 and M5 as monomers (feed ratio, 2.1:1) in yield of 47% or with M3 and M6 as monomers (feed ratio, 1:5) in yield of 87%, respectively [34]. With the optimum feed ratio in theory, the BFY2 was obtained in a lower yield. The reason for this phenomenon was that the bromine atoms in different monomers disrupt the sequential bond-forming process and cause additional competitive side reactions [35,36]. The presence of this phenomenon posed a disadvantage to the polymerization reaction, resulting in the production of LCPs with both low molecular weight and structural defects. Based on the above scenario, initial attempts to directly react M3 with M5 in THF/H_2_O mixture resulted in the formation of LCPs1 with low molecular weight (Figure 2, 2100 g/mol, *M*_w_ = 3400 g/mol, PDI = 1.62, Supporting Information, Appendix A).

To control the sequential bond-forming process in the tandem reaction and to inhibit the additional competitive side reactions, polymeric monomers should be chosen carefully. Herein, considering the order of halogen reactivity (I > Br) in Suzuki coupling, the polymeric monomeric were optimized by introducing iodine (I) into the structure of polymeric monomers. As shown in Appendix A, compounds M2 and M4 were prepared via lithium addition reaction and β-elimination reaction (Supporting Information). The model compound BFY2 was also synthesized via Pd(PPh_3_)_4_-catalyzed coupling reaction with M2 and M5 as monomers (feed ratio, 2:1) in yield of 90% or with M4 and M6 as monomers (feed ratio, 1:2.2) in yield of 93%, respectively. The synthesized BFY2 was confirmed by ^1^H NMR, ^13^C NMR and MS spectra, which was consistent with previous work [34]. The high efficiency and specificity of the model compound BFY2 indicated that the introduction of iodine (I) for a tandem reaction can effectively control the sequential bond-forming process and inhibit the additional competitive side reactions between Suzuki polymerization/Heck cyclization.

To further gain insight into the mechanistic pathway of the tandem reaction, we first performed a control experiment using M2 and M7 to test the Heck reaction (Figure 3). As shown in Appendix A, compound M7 was prepared via lithium addition reaction and β-elimination reaction (Supporting Information). The experiment (M2 as monomer) indicated that no new compound was obtained under the normal Suzuki condition; however, the intermediate M7 in Heck coupling could be smoothly converted into the target product BFY1 under normal Suzuki conditions in yield of 96%. These results supported that the tandem reaction probably first undergoes a Suzuki reaction and then a Heck reaction.

Then, we contributed our efforts toward further developing the above reaction for LCPs. The full ladder-type conjugated polymer LCPs2 was prepared via the Pd(PPh_3_)_4_-catalyzed tandem Suzuki polymerization/Heck cyclization reaction with M4 and M5 as monomers (Figure 2). The *M*_n_ of LCPs2 was determined to 9700 g/mol (Supporting Information, *M*_w_ = 16,600 g/mol, PDI = 1.71, Appendix A), which was almost 4-fold than that of LCPs1. LCPs2 was soluble in common organic solvents, such as THF, chloroform, toluene and N,N-dimethylformamide (DMF). The structures of M7, BFY1 and LCPs2 were confirmed by FTIR spectra (Figure 1). The spectrum of compound M7 showed a specific C-Br absorption peak (615 and 1030 cm^−1^) [37,38] and an =C-H absorption peak (1005 cm^−1^) that barely disappeared in the spectra of BFY1 and LCPs2, manifesting a high degree of Heck cyclization. Elemental analysis of LCPs2 revealed that the Br content was below 1%. The relatively lower weight percentage of Br value presumably resulted from the presence of terminal bromine groups. All these characterizations strongly supported the relatively high degree of Heck cyclization and the formation of LCPs with scarcely structural defects during the tandem reaction. The thermal stability studied by the thermogravimetric analysis (TGA) showed that the decomposition temperatures (5% weight loss) of the LCPs1 and LCPs2 were about 308 °C and 291 °C (Appendix A), respectively. Furthermore, we used differential scanning calorimetry (DSC) to study phase transition behaviors of LCPs1 and LCPs2, but no thermal transition (such as melting, crystallization or glass transition) on the DSC curves in the range of the measured temperature (ranged from −80 °C to 240 °C) was observed (Appendix A), indicating both LCPs1 and LCPs2 were amorphous in structure.

The normalized solution ultraviolet (UV) spectra were shown in Figure 2. Compared to BFY2, a redshift absorption of LCPs2 was observed. The optical bandgap (*E*_g_^opt^) of BFY2 and LCPs2 estimated from the onset of absorption band in CH_2_Cl_2_ solution were determined to be 1.97 eV and 1.51 eV, respectively. The electrochemical behaviors of BFY2 and LCPs2 in CH_2_Cl_2_ solution were investigated by cyclic voltammetry (Figure 3). The onset of the first reduction (*E*_re_^onset^) of BFY2 and LCPs2 were −1.21 and −1.07 V vs. Ag/AgNO_3_, respectively. The onset of the first oxidation (*E*_ox_^onset^) of BFY2 and LCPs2 were 0.63 and −0.54 V vs. Ag/AgNO_3_, respectively. The highest occupied molecular orbital/lowest unoccupied molecular orbital (HOMO/LUMO) values of BFY2 and LCPs2 were −5.18/−3.34 eV and −5.04/−3.48 eV, respectively.

## 4. Conclusions

In conclusion, the first Pd-catalyzed tandem Suzuki polymerization/Heck cyclization reaction for the fully ladder-type conjugated polymer has been developed. A few competitive side reactions between Suzuki polymerization/Heck cyclization were overcome with the introduction of iodine (I). LCPs2 has been successfully synthesized and the number-average molecular weight of LCPs2 was 9700 g/mol. The results demonstrated that this strategy is a feasible and facile approach to the construction of more LCPs.

## Data Availability

The data that support the findings of this study are available from the corresponding author.

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
