# Peer review of "Tandem Suzuki Polymerization/Heck Cyclization Reaction to Form Ladder-Type 9,9′-Bifluorenylidene-Based Conjugated Polymer"

_polymers, 2023, doi:10.3390/polym15163360_

Round 1

Reviewer 1 Report

The paper describes the synthesis of ladder-type 9,9’-bifluorenylidene-based conjugated polymer using tandem Suzuki polymerization/Heck cyclization reactions. The synthesis of materials is interesting but their characterization is very poor.

In my opinion, the quality of the paper is to low to be published in Polymers. In particular:

1.       The authors should clarify the reason of the synthesis and studies. The characterization of polymers says almost nothing about their applications. Why authors designed such structure of polymers?

2.       For the CV measurements authors said that “All potentials are depicted against the Fc/Fc+ redox couple.” But in the graph the x axis is described as “E/V Ag/AgNO3”, so what was the value of half-wave potential of Fc/Fc+ redox couple?

3.       Electrochemical properties should be described in the manuscript.

4.       The details for HOMO/LUMO calculations should be added, e.g. equations, the values of Epa onset, Epc onset, E1/2 ferrocene etc.

5.       The spectroscopic characterization of obtained compounds should be completed. The 1H NMR, 13C NMR, HRMS spectra and elemental analysis should be given for all compounds described in SI.

6.       Similarly, the data (1H NMR and elemental analysis) should be also added for polymer LCPs1.

7.       The equipment and measurement procedure for GPC,TGA and DCS should be added.

8.       Which standard was used to calibrate the instrument for GPC measurements?

9.       Why the TGA and DSC measurements were done only for LCPs2? What about LCPs1?

10.   The fluorene-based compounds are very often fluorescent. Did the compounds exhibited any emission properties?

Author Response

Reviewer: 1

Comments:

The paper describes the synthesis of ladder-type 9,9’-bifluorenylidene-based conjugated polymer using tandem Suzuki polymerization/Heck cyclization reactions. The synthesis of materials is interesting but their characterization is very poor.

In my opinion, the quality of the paper is to low to be published in Polymers. In particular:

  1. The authors should clarify the reason of the synthesis and studies. The characterization of polymers says almost nothing about their applications. Why authors designed such structure of polymers?

Reply: Thanks for your suggestions. We added some explanation in the manuscript.

“9,9’-bifluorenylidene (99’BF), as one of fulvalene derivatives, is a versatile scaffold with twelve different sits for functionalization by substitution. Though, 99’BF-based compounds has been used as optoelectronic materials, such as electron acceptor in bulky heterojunction solar cells, hole-transporting materials for perovskite solar cell and so on,[26-30] the synthesis of 99'BF-based compounds require extensive research and improvement due to its inherently crowded structure, which often results in low yield.[31-33] Meanwhile, the ladder-type 9,9’-bifluorenylidene-based conjugated polymers have not been reported.”

  1. For the CV measurements authors said that “All potentials are depicted against the Fc/Fc+ redox couple.” But in the graph the x axis is described as “E/V Ag/AgNO3”, so what was the value of half-wave potential of Fc/Fc+ redox couple?

Reply: Fc = (Eoc+Ere)/2 = (0.45 + 0.05)/2 = 0.25 eV (Figure R1)

Figure R1. The Cyclic voltammogram of Fc/Fc+ in CH2Cl2 solution.

  1. Electrochemical properties should be described in the manuscript.

Reply: Thanks for your suggestions. Electrochemical properties have been described in the manuscript. “The electrochemical behaviors of BFY2 and LCPs2 in CH2Cl2 solution were investigated by cyclic voltammetry (Figure 3). The onset of the first reduction (Ereonset) of BFY2 and LCPs2 were -1.21 and -1.07 V vs. Ag/AgNO3, respectively. The onset of the first oxidation (Eoxonset) of BFY2 and LCPs2 were 0.63 and -0.54 V vs. Ag/AgNO3, respectively. The highest occupied molecular orbital/lowest unoccupied molecular orbital (HOMO/LUMO) values of BFY2 and LCPs2 were -5.18/-3.34 eV and –5.04/-3.48 eV.”

  1. The details for HOMO/LUMO calculations should be added, e.g. equations, the values of Epa onset, Epc onset, E1/2 ferrocene etc.

Reply: HOMO = - (Eox(onset)-Fc+4.8) eV

LUMO = - (Ere(onset)-Fc+4.8) eV

EgCV = LUMO – HOMO

Egopt = 1240 / λ

Fc = (Eoc+Ere)/2 = (0.45 + 0.05)/2 = 0.25 eV

Table R1. The CV data of BFY2 and LCPs2

Eox(onset)

HOMO

Ere(onset)

LUMO

BFY2

0.63

-5.18

-1.21

-3.34

LCPs2

0.54

-5.04

-1.07

-3.48

  1. The spectroscopic characterization of obtained compounds should be completed. The 1H NMR, 13C NMR, HRMS spectra and elemental analysis should be given for all compounds described in SI.

Reply: Thanks for your suggestions. The 1H NMR, 13C NMR, HRMS spectra and elemental analysis had been given in Supporting Information (Figure S7, S8, S22 and S23).

  1. Similarly, the data (1H NMR and elemental analysis) should be also added for polymer LCPs1.

Reply: A good suggestion. The data (1H NMR, 13C NMR and elemental analysis) were added for LCPs1. (Supporting Information: Figure S22 and Figure S23. 1H NMR (CDCl3, 400 MHz): δ (ppm) 8.80-8.17 (broad, 3H), 7.84-7.30 (broad, 10H), 7.26-7.00 (broad, 3H), 1.56-0.85 (broad, 36H). 13C NMR (100 MHz, CDCl3) δ (ppm): 150.2, 149.8, 139.3, 139.0, 138.6, 137.1, 136.3, 135.5, 126.6, 126.4, 123.1, 122.7, 121.5, 119.4, 119.2, 117.5, 35.1, 34.9, 31.7, 31.5, 22.7, 14.2. Anal. calcd for C56H52: C 92.77, H 7.23; found: C 84.19, H 6.03, Br 3.26.)

  1. The equipment and measurement procedure for GPC,TGA and DCS should be added.

Reply: Thanks for your suggestions. The equipment and measurement procedure for GPC, TGA and DSC were added.

Supporting Information: Thermogravimetric analysis (TGA) was carried out at a heating rate of 10 °C/min from 40 °C to 800 °C under a nitrogen flow using a Q50 (TA, USA). Differential scanning calorimetry (DSC) measurement was performed at the rate of 10 °C/min from -80 °C to 240 °C under a nitrogen flow using a Q2000 (TA, USA). Number average molecular weight, weight average molecular weight and polydispersity index of polymers were measured by Shimadzu LC-20AT gel permeation chromatograph (GPC) using a PLgel 5 μm Mixed-D chromatographic column and a Shimadzu RID-20A differential refractometer. Tetrahydrofuran was used as eluent at a flow rate of 1.0 mL/min at 30 °C, and universal calibration was performed with standard polystyrene samples.

  1. Which standard was used to calibrate the instrument for GPC measurements?

Reply: Thanks for your suggestions. Indeed, the instrument for GPC measurements were calibrated by Polystyrene with different molecular weight (From Sigma-Aldrich. Product Number: 76552, Figure R2)

Figure R2. The GPC of Polystyrene with different molecular weight (SD1 and SD2).

  1. Why the TGA and DSC measurements were done only for LCPs2? What about LCPs1?

Reply: A good suggestion. We added the TGA and DSC measurements for LCPs1 (Figure R3 and Supporting information Figure S2).“The thermal stability studied by the thermogravimetric analysis (TGA) showed that the decomposition temperatures (5% weight loss) of the LCPs1 and LCPs2 was about 308 oC and 291 oC (Figure S2a). Furthermore, we used differential scanning calorimetry (DSC) to study phase transition behaviours, but no thermal transition in the range of the measured temperature was observed (Figure S2b), indicating both of LCPs1 and LCPs2 were amorphous structure.”

Figure R3. (a) DSC (b) TGA of LCPs1 and LCPs2.

  1. The fluorene-based compounds are very often fluorescent. Did the compounds exhibited any emission properties?

Reply: The compound and polymer did not exhibit emission properties.

Reviewer 2 Report

While reading the text, a number of questions arose.

1. How do the authors explain the presence of two symmetrical quartets of protons signals of tert-butyl fragments in 1H NMR? And why does their appearance change so much when moving from a monomeric compound to a polymeric one, Fig S6, S8, S14, S17 and S20? It would be logical to expect one triplet (or singlet), which is shown in the figure S17.

2. It is not entirely clear why the authors limited themselves to only one size exclusion chromatography and did not try to make mass spectra of the obtained compounds? Hence, there is no information on the degree of polymerization of the resulting macromolecules. I would like to recommend the use of time-of-flight matrix mass spectrometry, especially since the substances seem to be highly soluble in volatile solvents.

3. The performed chromatographic measurements also cause a certain kind of doubt, Figure S1. GPC of LCPs1 and LCPs2. Why do the detector readings have such a strong dip? Also, one of the compounds is represented by several maxima and I would like to take a look at the full version of the chromatogram.

4. From the text of the manuscript, I never got any idea about the phase behavior of the samples. Is it amorphous? There are no phase transitions on the DSC curves indicating melting, crystallization or glass transition. I advise the authors to take measurements from the negative area, for example, start with -120оС to 200 оC.

I believe that the manuscript can be published in the Journal Polymers after improvement.

Recommendation: Major revision.

Author Response

Reviewer: 2

Comments:

I believe that the manuscript can be published in the Journal Polymers after improvement.

Recommendation: Major revision.

While reading the text, a number of questions arose.

  1. How do the authors explain the presence of two symmetrical quartets of protons signals of tert-butyl fragments in 1H NMR? And why does their appearance change so much when moving from a monomeric compound to a polymeric one, Fig S6, S8, S14, S17 and S20? It would be logical to expect one triplet (or singlet), which is shown in the figure S17.

Reply: A good suggestion. The chemical environment of tert-butyl fragments on compounds M4, M7 and BFY2 are different, so they show two symmetrical quartets of protons signals, while the tert-butyl group of the compound BFY1 is in the same chemical environment, so it shows the single NMR peaks. Regarding the polymer LCPs2, it is important to note that LCPs2 actually consists of mixtures, wherein the chemical environment of tert-butyl fragments differs significantly.

  1. It is not entirely clear why the authors limited themselves to only one size exclusion chromatography and did not try to make mass spectra of the obtained compounds? Hence, there is no information on the degree of polymerization of the resulting macromolecules. I would like to recommend the use of time-of-flight matrix mass spectrometry, especially since the substances seem to be highly soluble in volatile solvents.

Reply: Thanks for your suggestions. Mass spectrometry was attempted for testing the degree of polymerization, but failed. However, the degree of polymerization was detected by using UV-vis absorption spectrum (Figure R4). The UV-vis absorption spectrum of LCPs1 exhibited two absorption peaks about 505 nm and 588 nm, range from 400-800 nm, revealing the low degree of polymerization. LCPs2 exhibits a broad absorption peak in the range of 400-800 nm, indicating the higher polymerization degree than LCPs1.

Figure R4. Normalized UV-vis absorption spectra of LCPs1 and LCPs2 in CH2Cl2 solution.

  1. The performed chromatographic measurements also cause a certain kind of doubt, Figure S1. GPC of LCPs1 and LCPs2. Why do the detector readings have such a strong dip? Also, one of the compounds is represented by several maxima and I would like to take a look at the full version of the chromatogram.

Reply: Thanks for your suggestions. Indeed, the Figure S1 had been exhibited the full version of the chromatogram. The number average molecular weight, weight average molecular weight and polydispersity index of polymers were measured by Shimadzu LC-20AT gel permeation chromatograph (GPC) using a PLgel 5 μm Mixed-D chromatographic column and a Shimadzu RID-20A differential refractometer. Tetrahydrofuran was used as eluent at a flow rate of 1.0 mL/min at 30 °C, and universal calibration was performed with standard polystyrene samples. After the 11 minutes, the test curve of the chromatogram is a straight line without any change.

  1. From the text of the manuscript, I never got any idea about the phase behavior of the samples. Is it amorphous? There are no phase transitions on the DSC curves indicating melting, crystallization or glass transition. I advise the authors to take measurements from the negative area, for example, start with -120 оС to 200 оC.

Reply: A good suggestion. The temperature range for the DSC test has been modified (-80 оС to 240 оС, Figure R5). No phase transitions were observed on the DSC curves, indication both LCPs1 and LCPs2 own the amorphous structure.

The lowest test temperature of the DSC instrument in our laboratory is -80 оС.

Figure R5. DSC of LCPs1 and LCPs2.

Round 2

Reviewer 1 Report

The authors have now revised the manuscript based on the concerns raised and the present version of the manuscript can be accepted for publication.